# Cryptic genetic variation shapes the adaptive evolutionary potential of enzymes

**Florian Baier[1], Nansook Hong[2], Gloria Yang[1], Anna Pabis[3], Charlotte M Miton[1], Alexandre Barrozo[3], Paul D Carr[2], Shina CL Kamerlin[3], Colin J Jackson[2], Nobuhiko Tokuriki[1]***

[1]Michael Smith Laboratory, University of British Columbia, Vancouver, Canada; [2]Research School of Chemistry, Australian National University, Canberra, Australia; [3]Department of Cell and Molecular Biology, Uppsala University, Uppsala, Sweden

**Abstract** Genetic variation among orthologous proteins can cause cryptic phenotypic properties that only manifest in changing environments. Such variation may impact the evolvability of proteins, but the underlying molecular basis remains unclear. Here, we performed comparative directed evolution of four orthologous metallo-β-lactamases toward a new function and found that different starting genotypes evolved to distinct evolutionary outcomes. Despite a low initial fitness, one ortholog reached a significantly higher fitness plateau than its counterparts, via increasing catalytic activity. By contrast, the ortholog with the highest initial activity evolved to a less-optimal and phenotypically distinct outcome through changes in expression, oligomerization and activity. We show how cryptic molecular properties and conformational variation of active site residues in the initial genotypes cause epistasis, that could lead to distinct evolutionary outcomes. Our work highlights the importance of understanding the molecular details that connect genetic variation to protein function to improve the prediction of protein evolution.

DOI: https://doi.org/10.7554/eLife.40789.001

*For correspondence:
tokuriki@msl.ubc.ca

**Competing interests:** The authors declare that no competing interests exist.

## Introduction

Genetic diversity across orthologous proteins is thought to be predominantly neutral with respect to their native, physiological function, but can cause variation in other non-physiological phenotypic properties, a phenomenon known as 'cryptic genetic variation' (*Le Rouzic and Carlborg, 2008*; *Gibson and Dworkin, 2004*; *Paaby and Rockman, 2014*). Cryptic genetic variation has been shown to play an important role in evolution because genetically diverse populations are more likely to contain genotypes with a 'pre-adapted' phenotype, for example, a latent promiscuous function, that may confer an immediate selective advantage when the environment changes and a new selection pressure emerges (*Queitsch et al., 2002*; *Specchia et al., 2010*; *de Visser et al., 2011*; *Amitai et al., 2007*; *Bloom et al., 2007*; *Rohner et al., 2013*). Beyond that specific scenario, however, we have little understanding of how genetic variation affects the long-term adaptive evolutionary potential (or 'evolvability') of proteins (*Wagner, 2008*). Many biological traits, in particular promiscuous protein functions, evolve by accumulating multiple adaptive mutations before reaching a fitness plateau or peak. Thus, the degree to which a trait can improve and the level of fitness that it can reach ultimately determines the evolutionary potential and outcome of a given genotype (*Romero and Arnold, 2009*). Recent studies have shown that intramolecular epistasis is prevalent and that the phenotypic effect of the same mutation can be drastically altered when introduced into a genotype that differs by only a few other mutations, which may lead to different evolutionary outcomes depending on the starting genotype (*Blount et al., 2008*; *de Visser and Krug, 2014*;

*Parera and Martinez, 2014*; *Starr and Thornton, 2016*; *Miton and Tokuriki, 2016*; *Harms and Thornton, 2014*; *Starr et al., 2017*). It has also been shown that orthologous enzymes can exhibit a broad range of phenotypic responses to the same selection pressure to improve their promiscuous activity (*Khanal et al., 2015*). These studies, however, have only focused on the effect of small mutational steps; the degree to which genetic starting points determines long-term evolutionary outcomes under identical selection remains unclear (*O'Loughlin, 2006*; *Burch and Chao, 2000*). More importantly, our understanding of the molecular mechanisms that underlie the relationship between genetic variation and evolvability is highly limited. It has been suggested that protein fold (*Tóth-Petróczy and Tawfik, 2014*; *Yip and Matsumura, 2013*; *Schulenburg et al., 2015*) and protein stability (*Weinreich and Chao, 2005*; *Bloom and Arnold, 2009*; *Tokuriki and Tawfik, 2009*) can shape evolvability, but these alone cannot explain the prevalence of epistasis and the enormous variation we observe in the evolvability of different genotypes. An in-depth understanding of the molecular constraints exerted by cryptic genetic variation is crucial to further develop our ability to decipher, predict and design the evolution of proteins.

Metallo-β-lactamases (MBLs) encompass a genetically diverse enzyme family that confer β-lactam antibiotic resistance to bacteria (*Figure 1A,B*) (*Meini et al., 2014*; *Bebrone, 2007*). Our previous work demonstrated that most MBLs exhibit phosphonate monoester hydrolase (PMH) activity ($k_{cat}/K_M$) in the range of 0.1 to 10 $M^{-1}s^{-1}$ (*Baier and Tokuriki, 2014*; *Baier et al., 2015*). β-lactam and phosphonate monoester hydrolyses differ in the cleaved bond (C-N *vs.* P-O) and the geometry of the transition state (tetrahedral *vs.* trigonal bipyramidal), making the PMH activity a catalytically distinct promiscuous activity for these enzymes. Thus, the evolution of MBLs toward PMH activity can be considered a model for neo-functionalization or the evolutionary emergence of a new enzyme function. Here, we carried out an empirical test of enzyme evolvability by performing directed evolution with four orthologous MBLs (NDM1, VIM2, VIM7, and EBL1) toward PMH activity. Employing diverse genetic, biochemical and biophysical analyses, we investigated the genetic and phenotypic response of MBLs to a new selection pressure during laboratory evolution. Consequently, we describe the molecular basis underlying these functional transitions: epistasis upon the fixation of a key initial adaptive mutation appears to differentiate the evolutionary trajectories. Last, we discuss the potential role of genetic variation in the evolution of new enzyme functions.

## Results

### Comparative directed evolution of four orthologous MBLs

We performed directed evolution using four different MBL orthologs, New Delhi metallo- β-lactamase 1 (NDM1), Verona imipenemase 2 and 7 (VIM2 and VIM7), and *Erythrobacter* β-lactamase 1 (EBL1), as starting genotypes (*Figure 1*, *Figure 1—figure supplement 1* and *Supplementary file 1A*). VIM2 and NDM1 were chosen due to the availability of their biochemical and structural information (*Garcia-Saez et al., 2008*; *King et al., 2012*). VIM7 and EBL1 are close homologous enzymes to VIM2 (80% amino acid sequence identity) and NDM1 (55% amino acid sequence identity), respectively (*Figure 1—figure supplement 1*). Although the catalytic efficiencies for the native β-lactams hydrolysis are similar across these enzymes ($k_{cat}/K_M$ only varies up to twofold across all six MBLS), their $k_{cat}/K_M$ for the promiscuous PMH activity varies by ~20 fold (*Figure 1A* and *Supplementary file 1B*). From these starting points, the same directed evolution scheme was applied to improve their ability to hydrolyze PMH in *Escherichia coli* (*Figure 1C*). Briefly, randomly mutagenized gene pools were transformed into *E. coli*, and purifying selection was used to enrich for functional variants while purging out non-functional variants by plating the libraries onto agar plates containing a low ampicillin concentration (4 μg/ml). Colonies were inoculated into liquid media (96-well plates, total of 396 variants per round), regrown, lysed and screened for PMH activity. The most improved variant(s) were isolated, sequenced and used as templates for the next round of evolution. Note that the 'PMH fitness' (i.e. the selection criteria) in our directed evolution scheme is defined as the level of PMH activity in *E. coli* cell lysate; as a result, PMH fitness is a function of catalytic activity and soluble expression levels in the cell. For example, the PMH fitness differences between NDM1 and VIM2 wild-type originate from their kinetic parameters, while their solubility is comparable (*Figure 1A,B*). By contrast, EBL1 and VIM2 wild-type have similar catalytic efficiencies

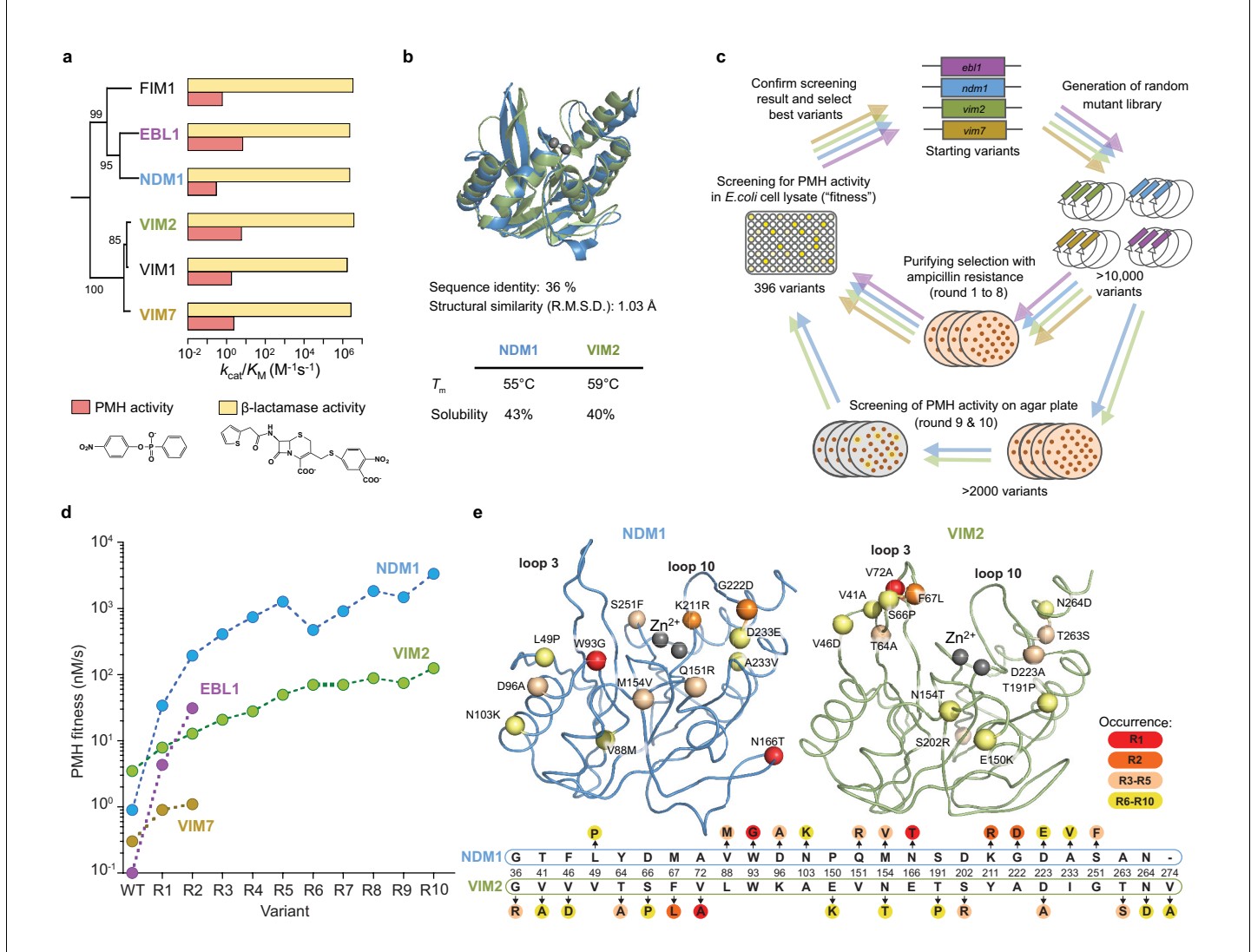

**Figure 1.** Comparative directed evolution of metallo-β-lactamases (MBLs). (A) Catalytic efficiencies ($k_{cat}/K_M$) of six metallo-β-lactamases (MBLs) for β-lactamase and PMH activities. Data are presented in *Supplementary file 1B*. The phylogenetic relationships are shown with bootstrap values indicated at each node. (B) Comparison of the overall 3D structures and biophysical properties of NDM1 (blue, PDB ID: 3SPU) and VIM2 (green, PDB ID: 1KO3). (C) Overview of the comparative directed evolution experiment of MBLs toward PMH activity. (D) PMH fitness (PMH activity in cell lysate) improvements of four MBLS during directed evolution. Data are presented in *Supplementary file 1G*. (E) The mutations fixed in NDM1 and VIM2 over ten rounds of directed evolution. The structural location of the mutations is mapped on the wild-type structures of NDM1 and VIM2, respectively, with the C-α atoms of the mutated residues shown as spheres (upper panel). Mutations are presented on a partial alignment between NDM1 and VIM2 sequences (bottom panel). A full alignment of MBLs and the mutations of the individual variants are presented in *Figure 1—figure supplement 1* and *Supplementary file 1F*.

DOI: https://doi.org/10.7554/eLife.40789.002

The following figure supplement is available for figure 1:

**Figure supplement 1.** Sequence comparison between metallo-β-lactamases.

DOI: https://doi.org/10.7554/eLife.40789.003

but their PMH fitness differ by 35-fold due to large differences in protein expression (*Figure 1A* and *Figure 1—figure supplement 1C*).

Over the first two rounds of directed evolution (R2), we observed significant differences in the extent of fitness improvement among the four parallel trajectories (*Figure 1D*). Interestingly, improvements in PMH fitness did not correlate with the initial fitness level. For example, although

VIM2 is the variant with the highest initial activity, it only exhibited a 4-fold improvement in PMH fitness over the wild-type, and was surpassed by other, initially less fit variants such as NDM1 and EBL1, with a 210- and 310-fold improvement in PMH fitness, respectively. Overall, the close pairs of homologues EBL1/NDM1 and VIM2/VIM7, seem to follow similar trends in PMH fitness improvement. Consequently, we continued the directed evolution of VIM2 and NDM1 only, for eight additional rounds (*Figure 1D* and *Supplementary file 1*). Overall, both trajectories exhibited diminishing returns in their evolution toward a new activity, that is each trajectory eventually reached a plateau where no substantial fitness increase was obtained in the last rounds of directed evolution. Note that in the last two rounds (R9 and R10), purifying selection was replaced by a pre-screening for increased PMH fitness (PMH activity level in the cell) on agar plates (>5000 variants), yet no variant with increased PMH fitness was isolated, suggesting that fitness plateaued in both trajectories (*Figure 1D*). Besides the similar overall trend along the trajectories, there are substantial differences in their evolutionary outcomes: while NDM1-WT was initially ~4 fold less-fit than VIM2-WT in PMH fitness, NDM1 fitness became ~28 fold higher than that of VIM2 by the end of the evolutionary experiment (R10). Overall, the NDM1 trajectory displays a ~ 3600 fold increase in PMH fitness (between NDM1-R10 and NDM1-WT), whereas the VIM2 trajectory only exhibits a ~ 35 fold increase in PMH fitness (*Figure 1D* and *Supplementary file 1G*), resulting in over a 100-fold difference in PMH fitness improvement between the two trajectories. Given that the two wild-type enzymes were almost identical in terms of their physicochemical properties, protein solubility, stability and structure (*Figure 1A,B*), the variation in evolutionary potential that separates these orthologous sequences is substantial.

## Genotypic solutions vary across the evolutionary trajectories

Overall, NDM1 and VIM2 accumulated 13 and 15 mutations, respectively. Interestingly, the mutations that occurred along each trajectory were entirely distinct (*Figure 1E* and *Supplementary file 1F*). Only two mutations occurred at the same position (154 and 223), and these were mutated to different amino acids. For example, the mutations in the early rounds of evolution, which conferred the largest PMH fitness improvements, were W93G, N116T, and K211R in the NDM1 trajectory, and V72A, and F67L for VIM2. In NDM1, most mutations are scattered relatively evenly around the active site; only one mutation (W93G) is located below loop 3, and several are located in and around loop 10 and other parts of the active site. By contrast, in VIM2, six mutations are tightly clustered within or next to loop 3. These results demonstrate that, for these two enzymes, distinct fitness outcomes result from distinct mutational responses.

## A limited accessibility of adaptive mutations from each starting point

An important question arising from the observation that NDM1 and VIM2 responded differently to the same selection pressure is whether their initial genotypes, and thus cryptic genetic variation, determined their evolutionary PMH fitness outcome. To address this, one should ideally employ experimental evolution using multiple parallel lineages and examine if evolution is repeatable from each starting point. However, performing such high-throughput characterizations is not feasible given our experimental evolution scheme. Nevertheless, we acquired several lines of evidence suggesting that starting genotypes influence the PMH fitness outcomes observed during MBL evolution (*Figure 2*).

First, we examined whether the adaptive mutations previously identified during directed evolution could be repeatedly selected by generating and screening three additional mutagenized libraries from each wild-type enzyme (i.e. replaying R1 evolution). Strikingly, most positive variants isolated in the additional screening possess mutations at the same positions as previously identified: Trp93 (in R1) and Met154 (R5) were isolated for NDM1. For VIM2, Val72 (R1), Phe67 (R2), Ser202 (R4), Thr263 (R5), and Val41 (R6) were isolated again (*Figure 2A*, *Figure 2—figure supplement 1* and *Supplementary file 1H*). These results suggest that only a handful of adaptive mutations, which are unique to each genotype, are available, at least from the wild-type sequence.

Next, we assessed the PMH fitness effects of the NDM1 and VIM2 mutations by introducing the mutations fixed during R2-R4 directly into their corresponding wild-type genetic background. In the VIM2 trajectory, we observed no differential fitness effect when later mutations are directly introduced on the wild-type background. This is consistent with the isolation of these later mutations

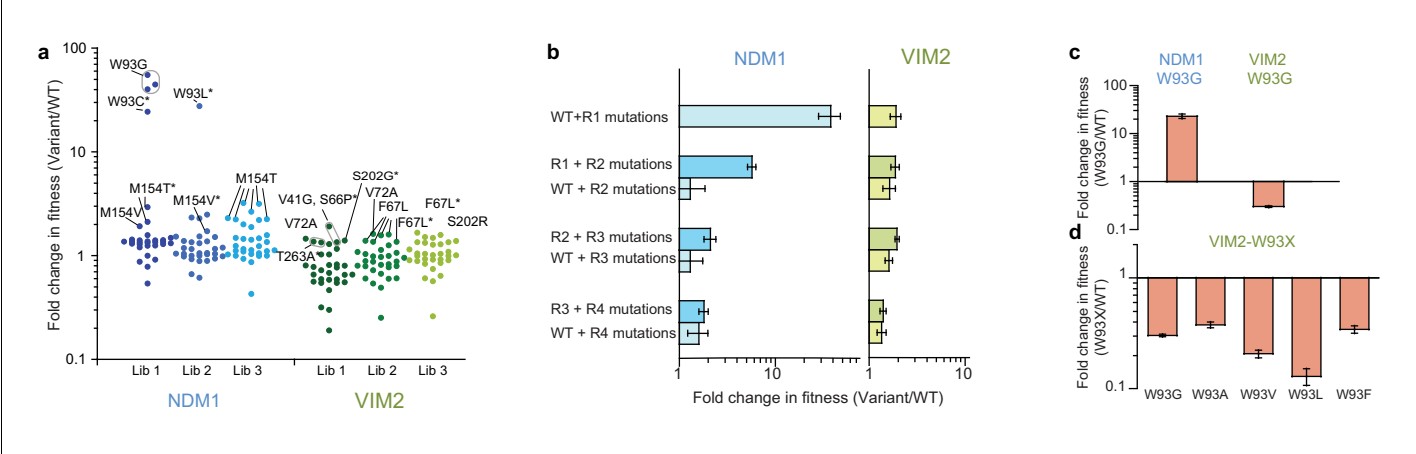

**Figure 2.** Accessibility of adaptive mutations from each starting point. (**A**) Additional screenings of NDM1 and VIM2 round one libraries. Activities of the top 30 variants isolated from additional screenings of three independently generated mutagenized libraries and normalized on the PMH fitness of NDM1-WT and VIM2-WT, respectively; presented as scatterplots. Mutants are labeled when variants possess a mutation previously encountered in the original directed evolution or if this mutation is located at the same position. Asterisks indicate that the variant has other mutations in addition to the labeled one. Mutations and activities of all improved variants are listed in *Supplementary file 1H*. The activities of all 2160 variants in the round one additional screenings are presented in *Figure 2—figure supplement 1*. (**B**) Epistasis analysis of mutations. Mutations occurring in the NDM1 and VIM2 trajectories were introduced into their respective wild-type sequence and the resulting PMH fitness was compared to the original PMH fitness change observed during the trajectory. (**C**) Fold change in PMH fitness induced by mutation W93G in NDM1-WT and VIM2-WT, respectively. (**D**) Fold change in PMH fitness induced by mutation of Trp93 in VIM2 to various hydrophobic residues. A full description of the lysate activities is presented in *Supplementary file 1G*. Errors bars represent the propagated standard deviation from triplicate measurements.
DOI: https://doi.org/10.7554/eLife.40789.004

The following figure supplements are available for figure 2:

**Figure supplement 1.** Screening of additional libraries.
DOI: https://doi.org/10.7554/eLife.40789.005

**Figure supplement 2.** Differential effects of mutations (W93G and V72A) on six metallo-β-lactamases.
DOI: https://doi.org/10.7554/eLife.40789.006

during the additional R1 screening experiment, suggesting that a range of adaptive mutations is readily accessible from the starting enzyme. As a consequence, these mutations can collectively improve PMH fitness, regardless of their order of appearance and without significant epistatic constraint. On the contrary in NDM1, later mutations only confer a selective advantage if introduced after the initial fixation of R1 mutations (*Figure 2B*). These results explain why only few mutations, for example W93G, were identified in the additional screening from the starting enzyme. This also suggests that W93G plays a permissive role for later mutations, and thus that the order of appearance of mutations in the NDM1 trajectory is rather constrained.

Last, to examine why distinct mutations arise in the first round of the NDM1 and VIM2 trajectories, we examined the effect of the initial mutation from each trajectory (W93G in NDM1 and V72A in VIM2) in their counterpart enzyme. We found that each trajectory's adaptive mutations are incompatible with one another. While W93G increases PMH fitness in cell lysate by ~25 fold in NDM1-WT (104-fold with purified NDM1), it causes a 3-fold decrease in PMH fitness when introduced into VIM2-WT (10-fold with purified VIM2), explaining why VIM2 never acquired this mutation (*Figure 2C, E* and *Figure 2—figure supplement 2*). W93G exhibits beneficial effects in MBL homologs other than VIM2 and VIM1, consistent with the observation that Trp93 was not only selected in the NDM1 trajectory, but also in two other evolutionary trajectories, EBL1 and VIM7's (W93L in EBL1 and W93G in VIM7, respectively, *Supplementary file 1F*). Moreover, introducing other hydrophobic residues (A, V, L and F) at position 93 in VIM2 had similar negative effects, indicating that mutating Trp93 is largely deleterious in VIM2 (*Figure 2D*). Similarly, V72A, which improved the PMH fitness of VIM2 by ~2 fold (1.7-fold with purified enzyme), was largely neutral in NDM1 (*Figure 2—figure supplement 2* and *Supplementary file 1I*).

## Distinct molecular changes underlie the two evolutionary trajectories

The conventional paradigm of protein evolution is dominated by the idea that higher protein stability, or greater soluble and functional expression, promotes protein evolvability, by buffering the destabilizing effect of function-altering mutations and allowing a greater number of adaptive mutations to accumulate (*Bloom and Arnold, 2009*; *Weinreich and Chao, 2005*; *Tokuriki and Tawfik, 2009*). This model, however, fails to predict the difference in evolvability between NDM1 and VIM2, because their relative stability and soluble expression are similar (*Figure 1B*). In order to elucidate the precise molecular changes that enabled evolutionary optimization in each trajectory, we measured a range of molecular properties, including catalytic efficiency ($k_{cat}/K_M$), the fraction of soluble protein expression, melting temperature ($T_m$), and oligomeric assembly, over the course of their evolution. The molecular changes that underlie their respective PMH fitness (PMH activity in cell lysate) improvements differed substantially (*Figure 3*, *Figure 3—figure supplement 1* and *Supplementary file 2*). By R10, the catalytic efficiency of NDM1 was improved by more than 18,000-fold compared to the wild-type ($k_{cat}/K_M$ from 0.32 to 5900 $M^{-1}s^{-1}$, *Figure 3A,B* and *Supplementary file 2B*). This significant improvement in catalytic efficiency, however, was offset by a loss of soluble expression at R1, where $k_{cat}/K_M$ increased by ~300 fold, while the solubility decreased from 43% to 25% (*Figure 3C,D*). The level of soluble expression never recovered, while $k_{cat}/K_M$ gradually increased until reaching the plateau observed in R7. By contrast, the catalytic efficiency of VIM2 stagnated at round 6, with only a ~ 25 fold increase in $k_{cat}/K_M$ overall (VIM2-R10 compared to the wild-type). The subsequent PMH fitness improvements resulted from increasing soluble expression, from 40% to 70% (*Figure 3*). Changes in solubility are only weakly correlated to changes in $T_m$ (*Figure 3—figure supplement 1*), indicating that other factors such as kinetic stability or protein folding affect the level of soluble protein expression more than thermostability (*Bershtein et al., 2012*; *Wyganowski et al., 2013*; *Lim et al., 2016*; *Pey et al., 2011*). NDM1 retains the same monomeric quaternary structure along its trajectory to NDM1-R10. The monomeric VIM2-WT, however, evolved to exist in an equilibrium between monomer and dimer by VIM2-R10 (*Figure 3—figure supplement 2A*). We isolated monomeric and dimeric states, respectively, using

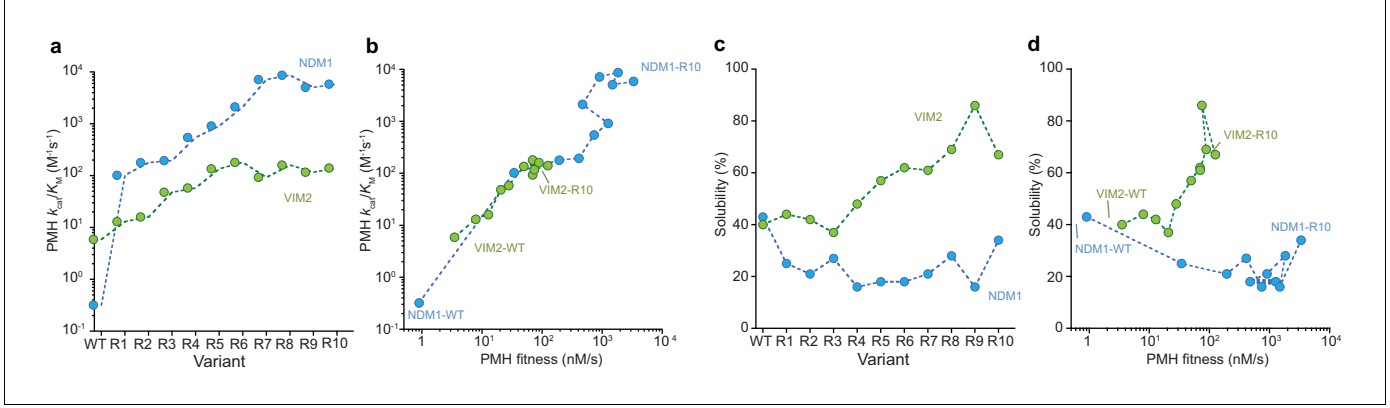

**Figure 3.** Molecular and phenotypic changes along the NDM1 and VIM2 evolutionary trajectories. (A) Catalytic efficiencies ($k_{cat}/K_M$) of purified variants for PMH activity. (B) Correlation between PMH fitness (cell lysate activity) and catalytic efficiency ($k_{cat}/K_M$ in $M^{-1}s^{-1}$) of purified enzymes along the evolutionary trajectories. (C) Changes in solubility during evolution. SDS-PAGE images of the soluble and insoluble protein expressions for each variant are presented in *Figure 3—figure supplement 1*. (D) Correlation between PMH fitness and solubility along the evolutionary trajectories. Individual catalytic, thermostability, and solubility parameters are listed in *Supplementary file 2*.

DOI: https://doi.org/10.7554/eLife.40789.007

The following figure supplements are available for figure 3:

**Figure supplement 1.** Solubility and melting temperature of variants.
DOI: https://doi.org/10.7554/eLife.40789.008

**Figure supplement 2.** Oligomerization states of NDM1 and VIM2.
DOI: https://doi.org/10.7554/eLife.40789.009

**Figure supplement 3.** Effect of expression temperature on cell lysate PMH fitness (cell lysate activity in nM/s).
DOI: https://doi.org/10.7554/eLife.40789.010

size-exclusion chromatography, and determined their kinetic parameters. The catalytic efficiency of the dimeric state for PMH is eightfold lower compared to the monomeric state ($k_{cat}/K_M$ = 510 and 60 $M^{-1}s^{-1}$ for the monomer *vs.* dimer, respectively), indicating that the dimer formation may be associated with an overall increase in solubility, rather than catalytic efficiency along the VIM2 trajectory (*Figure 3—figure supplement 2*) (*Bershtein et al., 2012*).

Altogether, our results suggest that two distinct molecular mechanisms underlie the observed differences in evolutionary trajectories. NDM1 adapted exclusively by increasing catalytic efficiency while compromising its soluble expression while VIM2 evolved by increasing its expression, oligomerization and activity. Our observations also indicate that differences in protein stability or soluble expression in the initial genotypes cannot explain the differences in evolvability between the two enzymes. VIM2 variants consistently exhibit higher stability and solubility throughout the trajectory, yet the improvements in $k_{cat}/K_M$ and PMH fitness are far lower than that of NDM1. Moreover, the distinct phenotypic solutions further highlight the qualitatively different evolutionary processes that led to distinct PMH fitness plateaus in each trajectory. Interestingly, the genetic differences in the evolutionary end points can create additional cryptic variation that becomes apparent when the environment is changed: when the enzymes are expressed at 37°C (rather that 30°C at which they evolved for higher PMH fitness) NDM1 variants are significantly less fit because of their lower soluble expression, whereas VIM2 variants maintain similar PMH fitness levels even at higher temperature (*Figure 3—figure supplement 3*).

## Distinct structural adaptation between the evolutionary trajectories

Having established that protein stability does not constrain the evolvability of the enzymes, we sought a molecular explanation by solving the crystal structures of the R10 variants of the NDM1 and VIM2 trajectories, allowing us to compare them with the previously published wild-type structures (*Figure 4* and *Supplementary file 3*) (*King et al., 2012*; *Garcia-Saez et al., 2008*). For NDM1-R10, we obtained crystal structures of the apo-enzyme and in complex with the phenylphosphonate product bound in the active site after *in crystallo* hydrolysis. Additionally, we conducted molecular dynamics (MD) simulations in complex with the PMH substrate (*p*-nitrophenyl phenylphosphonate, PPP). We identified three main structural adaptations from NDM1-WT that may underlie the ~18,000 fold improvement in catalytic efficiency for PMH activity. First, W93G removes steric hindrance between the side chain of Trp93 and the substrate, and generates a complementary pocket for the phenyl group below loop 3 (*Figure 4B* and *Figure 5—figure supplement 1*). Second, there is an inward displacement of loop 3 (Trp93 is located near the base of this loop) by ~6 Å, which allows for improved π-π stacking interactions between Phe70 and the *p*-nitrophenol-leaving group (*Figure 4C* and *Figure 5—figure supplement 1*). Third, loop 10 is repositioned via reorganization of the local hydrogen bonding network (largely by K211R and G222D), allowing it to interact with the leaving group of the substrate (*Figure 4C*). For VIM2, we were only able to crystallize the dimeric fraction of VIM2-R10. This dimer reveals an unprecedented structural rearrangement: a full half of the structure exhibits a symmetrical domain-swapping between the two subunits (*Figure 4D* and *Figure 4—figure supplement 1*). Besides domain swapping, the major structural rearrangement occurring between VIM2-WT and VIM2-R10 involves the reorganization of loop 3 resulting in its disorder in VIM2-R10, which is most likely caused by six mutations within and next to loop 3 (*Figures 1E* and *4D*). Four out of these six loop 3 mutations have accumulated by R5; thus, we speculate that these mutations and the loop rearrangement were the major cause of the ~25 fold increase in catalytic efficiency for PMH in VIM2-R10, compared to the wildtype. Taken together, the results further emphasize that the two starting enzymes responded to the same selection pressure by substantially different molecular solutions.

## Molecular basis for the mutational incompatibility of the key mutation W93G

Finally, we investigated the molecular basis underlying the functional effect of W93G, a key mutation that differentiates the two trajectories, by measuring the catalytic activities of the purified enzymes at single enzyme and substrate concentrations. In NDM1, mutation W93G causes ~100 fold increase in PMH activity compared to the wild-type, but when introduced into VIM2, it reduces its PMH activity by 10-fold (*Figure 2—figure supplement 2* and *Supplementary file 1I*). We performed and

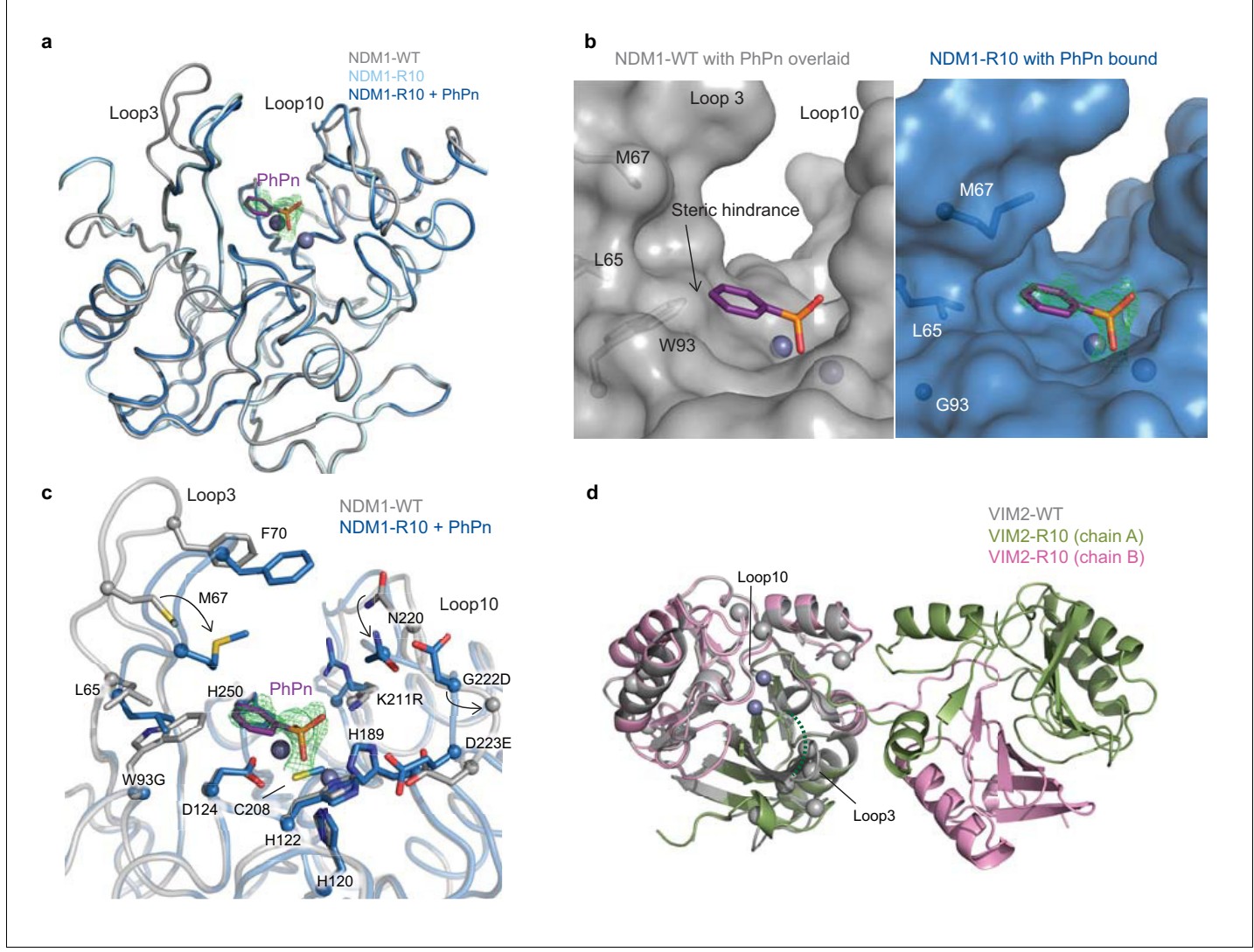

**Figure 4.** The structural basis for improved PMH activity. (A) Structural overlay of NDM1-WT (grey, PDB ID: 3SPU) and NDM1-R10 in the apo form (cyan, PDB ID: 5JQJ) and NDM1-R10 in complex with the PMH hydrolysis product, phenylphosphonate (PhPn, magenta sticks) (blue, PDB ID: 5K4M). The active site metal ions are shown as spheres. (B) Surface views of the active site of NDM1-WT and NDM1-R10 with the PMH product. The product in the NDM1-WT structure is overlaid from the NDM1-R10 structure. The mFo-dFc omit electron density for the phenylphosphonate is shown (green mesh), contoured at 2.5σ. (C) Comparison of active site residues between NDM1-WT and NDM1-R10. (D) Structural overlay of monomeric VIM2-WT (grey, PDB ID: 1KO3) and domain swapped-dimeric VIM2-R10 (green and pink, PDB-ID: 6BM9). The disordered loop three in VIM-R10 is indicated as a dashed green line. The mutations accumulated during the VIM2 trajectory are depicted as light grey spheres. The mFo-Fc omit electron density of the structure can be found in *Figure 4—figure supplement 1*.

DOI: https://doi.org/10.7554/eLife.40789.011

The following figure supplement is available for figure 4:

**Figure supplement 1.** Structure of the domain swapped-dimeric VIM2-R10.

DOI: https://doi.org/10.7554/eLife.40789.012

compared MD simulations of NDM1-WT, VIM2-WT and models of NDM1-W93G, and VIM2-W93G in the presence of the PMH substrate (*Figure 5*, *Figure 5—figure supplement 2* and *Supplementary file 4*). NDM1-W93G showed similar structural adaptations to NDM1-R10 (described above); W93G eliminates the steric hindrance with the substrate, and allows loop 3 to shift inward to promote complementary interactions with the substrate, without the rearrangement of loop 10, which presumably occurs later in evolution. By contrast, in VIM2-WT, Trp93 adopts a different orientation to avoid steric hindrance, which instead promotes complementary interactions with the phenyl

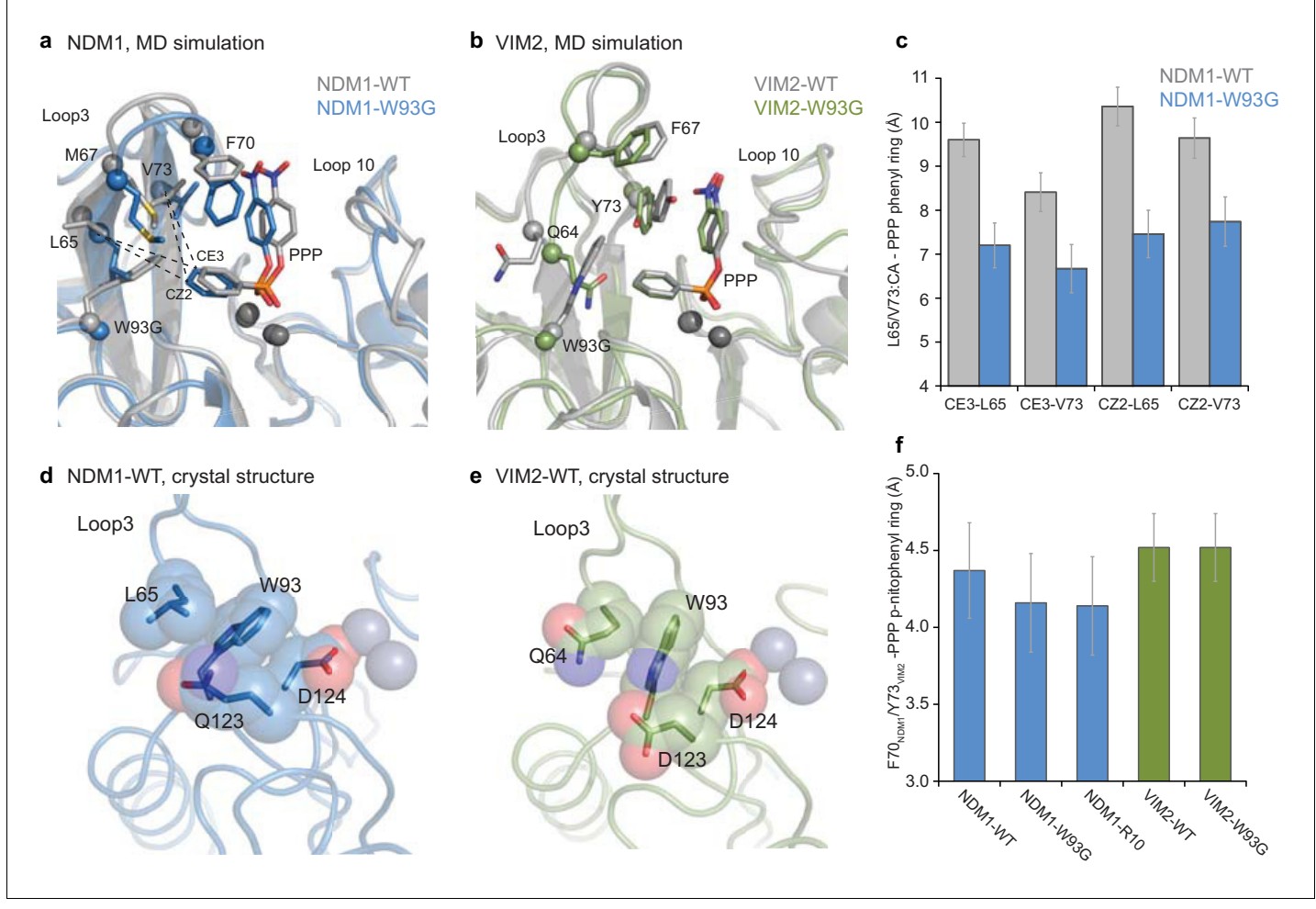

**Figure 5.** Comparisons of the structural changes induced by mutation W93G. Representative structures from MD simulations of (**A**) NDM1-WT (grey) and NDM1-W93G (blue), and (**B**) VIM2-WT (grey) and VIM2-W93G (green). (**C**) Changes in hydrophobic interactions between the phenyl ring of the PMH substrate (CE3 and CZ2 atoms) and residue Leu65 and Val73 ($\alpha$-carbon) of NDM1 upon W93G, during the MD simulations. (**D and E**) Comparison of the position of Trp93 and second shell residues in the (**D**) NDM1-WT (PDB-ID: 3SPU) and (**E**) VIM2-WT (PDB-ID: 1KO3) crystal structures. (**F**) Average distance between the *p*-nitrophenyl ring of the PMH substrate and the side chain rings of Phe70 in NDM1, and Tyr73 in VIM2, when the two rings form a $\pi$-$\pi$ interaction during the MD simulations.

DOI: https://doi.org/10.7554/eLife.40789.013

The following figure supplements are available for figure 5:

**Figure supplement 1.** The molecular dynamics (MD) simulations of NDM1-WT and NDM1-R10.
DOI: https://doi.org/10.7554/eLife.40789.014

**Figure supplement 2.** The molecular dynamics (MD) simulations of NDM1-WT, NDM1-W93G, VIM2-WT, and VIM2-W93G.
DOI: https://doi.org/10.7554/eLife.40789.015

ring of the substrate. Consequently, W93G in VIM2 removes beneficial substrate-enzyme interactions, thus causing a deleterious effect on catalytic efficiency. Similarly, unlike in NDM1, loop 3 in VIM2-W93G does not form complementary interactions with the substrate, which is consistent with the observation that loop 3 is extensively mutated and reorganized by other mutations later in the VIM2 trajectory. By examining these crystal structures, we found that the 'second-shell' residues of Trp93 cause its indole side chain to adopt a different orientation. In NDM1, Leu65, Gln123 and Asp124 constrain Trp93 to point toward the active site (*Figure 5D*). In VIM2, however, the second shell residues (Gln65, Asp123, and Asp124) differ, resulting in Trp93 being stabilized in an alternative conformation (*Figure 5E*). Thus, remote and seemingly neutral sequence variation between the enzymes has substantial impact on the mutagenesis of a key active site residue.

We expanded our mutational analysis of W93G and V72A (the first mutation in the VIM2 trajectory) by measuring their effects on four other orthologous enzymes: EBL1, Florence Imipenemase 1 (FIM1), VIM1, and VIM7; with purified enzymes at single substrate and enzymes concentrations. We found that their effects on both PMH and β-lactamase activities vary consistently and significantly, even between orthologs with high sequence identity (*Figure 2—figure supplement 2* and *Supplementary file 1I*). For example, in VIM7, which is 80% identical to VIM2, W93G caused a ~ 3 fold increase in PMH activity, despite the same mutation causing an antagonistic 10-fold decrease in PMH activity in VIM2. Taken together, cryptic and subtle differences in sequence and structure in both orthologs may influence the conformation of Trp93, a key active site residue, and cause the observed 1000-fold difference in the phenotypic effect of mutation W93G (100-fold increase in NDM1 *vs.* 10-fold decrease in VIM2). Thus, as previously discussed, the accessibility of adaptive mutations from each starting point, at least within one mutational step, seems to depend on the starting genotype (*Figure 2*).

## Discussion

The existence of a great diversity of protein functions, together with many contemporary examples of proteins that have promptly adapted to changing environments, suggest a remarkable degree of evolvability for biological molecules (*Wagner, 2008*; *Aharoni et al., 2005*; *Baier et al., 2016*). These successful examples of adaptation, however, may also obscure a wealth of cases where proteins have failed to adapt or were limited in their ability to evolve a new function. Our observations highlight that not all enzymes evolve to their fitness maximum, and that seemingly innocuous genetic variation may result in significant consequences for a protein's ability to evolve a new function. Metallo-β-lactamases (MBLs) confer β-lactam antibiotic resistance to bacteria by cleaving β-lactam rings (C-N bond). The PMH activity (breaking P-O bond) in MBLs is not likely to have ever been selected for; it can thus be considered as a truly non-physiological and serendipitous promiscuous function (*Khersonsky and Tawfik, 2010*; *Babtie et al., 2010*; *Copley, 2015*). Thus, the promiscuous PMH activity of MBLs can be considered a cryptic and hidden 'pre-adapted' property. Genetic variation can alter the level of promiscuous activities in enzymes (*de Visser et al., 2011*; *Bloom et al., 2007*; *Amitai et al., 2007*). Indeed, in MBLs, the catalytic efficiencies for PMH hydrolysis differ by ~20 fold among orthologous enzymes, while the catalytic efficiencies for β-lactams hydrolysis only vary by twofold (*Figure 1A*). Such phenotypic variation can cause differences in evolvability when an 'immediate' response to changing environments is essential (*de Visser et al., 2011*; *Bloom et al., 2007*; *Amitai et al., 2007*). Our work extends this view of the role of genetic diversity to encompass the evolutionary responses of individual genes to new selection pressures (*Khanal et al., 2015*). Importantly, the initial fitness and/or the catalytic efficiencies of enzymes do not necessarily provide a good indicator of their evolutionary outcome, and thus of their evolvability (*Khanal et al., 2015*). Genetic variation can significantly alter the accessibility of adaptive mutations, and starting points with lower fitness may reach much higher evolutionary plateaus than starting points endowed with higher initial fitness. Thus, adaptive evolutionary potential can be truly 'cryptic' and only apparent after evolution happens, further complicating the prediction of the most successful initial genotypes.

Our observations, when combined with those of others, indicate that the evolvability of proteins is generally and profoundly rooted in the starting genetic variation that is available. On one hand, examples from nature and the laboratory have shown that independent evolutionary trials from a single genotype often follow very similar genetic and thus phenotypic trajectories; suggesting that only a limited amount of beneficial mutations is available from any single genotypic starting point, and in those cases evolution is largely deterministic (*Weinreich et al., 2006*; *Salverda et al., 2011*; *Lobkovsky and Koonin, 2012*; *Dickinson et al., 2013*; *Palmer et al., 2015*; *Kaltenbach et al., 2015*). On the other hand, the prevalence of epistasis causes distinct effects when some adaptive mutation(s) are introduced on the genetic background of otherwise phenotypically similar orthologs (*Khanal et al., 2015*; *Parera and Martinez, 2014*; *Harms and Thornton, 2014*; *Starr et al., 2017*), suggesting that genetic variation epistatically 'restricts' and/or 'permits' the accessibility of certain adaptive mutations (*Harms and Thornton, 2013*; *Kaltenbach and Tokuriki, 2014*; *Starr and Thornton, 2016*). The successful evolution of new protein functions may therefore rely on genetic drift to explore the sequence space, thereby creating a diverse genotypic pool presenting multiple, differentially evolvable, genotypes from which an evolutionary trajectory may arise (*Harms and Thornton,*

*2014*). The genetic diversity that these neutral drifts generate thereby provides a foundation for the response of a population to new selection pressures (*Wagner, 2008*; *Harms and Thornton, 2013*). One complicating aspect of this model is the impact of temporal and spatial selection on the emergence of diverse genotypes. It is possible that this model explains a commonly observed phenomenon in bacterial adaptation, for example, drug-resistance and xenobiotic degradation (*Copley, 2009*; *Singh, 2009*), in which typically only a small number of genotypes endowed with new functions will emerge within a larger population, after which these successfully adapted genes quickly disseminate to other bacteria via horizontal gene transfer.

Our observations shed light on the molecular mechanisms underlying genetic variation and evolution of new enzymatic function. The established and conventional paradigm of protein evolution suggests that protein stability, or soluble and functional expression of proteins, is the dominant factor in determining evolvability (*Bloom and Arnold, 2009*; *Tokuriki and Tawfik, 2009*; *Weinreich and Chao, 2005*). Other studies indicate that protein structural folds cause differences in evolvability (*Yip and Matsumura, 2013*; *Schulenburg et al., 2015*; *Dellus-Gur et al., 2013*; *Tóth-Petróczy and Tawfik, 2014*). However, in our study, neither protein stability nor protein fold can account for variation in evolutionary potential between the MBL enzymes toward PMH activity. Instead, we found that cryptic and subtle molecular differences seem to have a greater impact on granting the accessibility of key initial adaptive mutations (such as W93G): although VIM2 and NDM1 possess an almost identical overall structure, thermostability and soluble expression level, their evolution resulted in substantially different fitness outcomes and molecular solutions. Indeed, many examples demonstrated that strong epistasis in protein function can be caused by subtle conformational changes through the accumulation of mutations (*Tomatis et al., 2008*; *Harms and Thornton, 2014*; *Dellus-Gur et al., 2015*; *Yang et al., 2016*; *Campbell et al., 2016*). Thus, an in-depth understanding of the molecular and evolutionary mechanisms underlying the optimization of enzymatic functions, in particular of promiscuous activities, is essential to developing our ability to understand and predict the evolution of enzymes.

These results also have implications for protein engineering, laboratory evolution and protein design. Protein engineers often choose a single starting genotype based on the availability of biochemical and structural information (often orthologs with the highest initial activity). While much effort has been devoted to developing technologies for better library construction and high-throughput screening (*Dalby, 2011*; *Davids et al., 2013*; *Goldsmith et al., 2017*), our observations suggest that it may also be effective to explore diverse initial genotypes and identify the most evolvable starting sequences to successfully obtain an optimized functional protein (*Lapidoth et al., 2018*; *Kiss et al., 2013*). Yet, predicting the evolvability of proteins remains a challenge that will require more in-depth biochemical and biophysical characterizations in order to dissect the molecular mechanisms driving the evolution of protein function (*Miton et al., 2018*; *Clifton et al., 2018*; *Kaltenbach et al., 2018*). The biophysical rationalization of the cryptic properties of MBL enzymes described here contributes to the ambitious goal of understanding the molecular mechanisms behind neutral genetic variation and evolvability that we observe in nature, in such a way as to allow us to predict evolutionary pathways and understand how to acquire better biological molecules.

## Materials and methods

### Generation of mutagenized library

Random mutant libraries were generated with error-prone PCR using nucleotide analogues (8-Oxo-2'-deoxyguanosine-5'-Triphosphate (8-oxo-dGTP) and 2'-Deoxy-P-nucleoside-5'-Triphosphate (dPTP); TriLink). Two independent PCR reactions were prepared, one with 8-oxo-dGTP and one with dPTP. Each 50 µl reaction contained 1 × GoTaq Buffer (Promega), 3 µM MgCl$_2$, 1 ng template DNA, 1 µM of primers (forward (T7 promoter): taatacgactcactataggg; reverse (T7 terminator): gctagttattgctcagcgg), 0.25 mM dNTPs, 1.25 U GoTaq DNA polymerase (Promega) and either 100 µM 8-oxo-dGTP or 1 µM dPTP. Cycling conditions: Initial denaturation at 95°C for 2 min followed by 20 cycles of denaturation (30 s, 95°C), annealing (60 s, 58°C) and extension (70 s, 72°C) and a final extension step at 72°C for 5 min. Subsequently, each PCR was treated with *Dpn* I (Fermentas) for 1 hr at 37°C to digest the template DNA. PCR products were purified using the Cycle Pure PCR purification kit (E.N.Z.A) and further amplified with a 2 × Master mix of Econo TAQ DNA polymerase

(Lucigen) using 10 ng of the template from each initial PCR and the same primers at 1 µM in a 50 µl reaction volume. Cycling conditions: Initial denaturation at 95°C for 2 min followed by 30 cycles of denaturation (30 s, 95°C), annealing (20 s, 58°C) and extension (70 s, 72°C) and a final extension step at 72°C for 2 min. The PCR products were purified and cloned as described above. Sequencing of naïve libraries for NDM1 and VIM2 revealed differences in mutation rates between the two genes. The variants exhibited on average 2.6 mutations/gene for NDM1 and 4.3 mutations/gene for VIM2 (*Supplementary file 1C*). Since the two genes were mutated under identical conditions, this is likely due to intrinsic sequence properties of the templates, in terms of DNA composition for example.

## Generation of DNA shuffling libraries

The staggered extension process (StEP) protocol is a DNA shuffling method that was used at rounds where multiple variants exhibited higher PMH fitness (rounds 2 and 5 in NDM1; 4 and 10 in VIM2), in order to enrich beneficial mutations and purge deleterious or neutral ones (*Supplementary file 1F*) (*Zhao et al., 1998*). Plasmids of variants were mixed in equimolar amounts to 500 ng of total DNA and used as a template for the StEP reaction. Cycling conditions: Initial denaturation at 95°C for 5 min followed by 100 cycles 95°C for 30 s followed by 58°C for 5 s. PCR products were purified using the Cycle Pure PCR purification kit and further amplified with a 2 × Master mix of Econo TAQ DNA polymerase. Libraries were cloned into pET29(b) as described above.

## Site-directed mutagenesis

Single-point mutant variants were constructed by site-directed mutagenesis as described in the QuickChange Site-Directed Mutagenesis manual (Agilent) using specific primers. All variants contained only the desired mutation, which was confirmed by DNA sequencing.

## Pre-screen on agar plates

Libraries in pET29-pMBP were transformed into *E. coli* BL21 (DE3) cells and incubated for 1 hr at 37°C prior to plating. For the low antibiotic prescreen, the transformants were plated on agar plates (150 mm diameter) containing 4 µg/ml ampicillin, 0.1 mM IPTG, 200 µM $ZnCl_2$ and 40 µg/ml kanamycin, yielding >500 colonies. The minimum inhibitory concentration (MIC) of ampicillin for *E.coli* cells expressing NDM1 and VIM2 is 256 µg/ml, whereas for the *E.coli* cells alone it is <2 µg/ml. Subsequently, surviving colonies were directly picked from plates for rescreening in 96-well plates. For the direct PMH prescreen, transformation reactions were plated on six agar plates (150 mm diameter) containing 40 µg/ml kanamycin, such that each plate contained between 400 and 2000 colonies. Colonies were replicated onto nitrocellulose membranes (BioTrace NT Pure Nitrocellulose Transfer Membrane 0.2 µm, PALL Life Sciences), which were subsequently placed onto LB agar plates containing 1 mM IPTG, 200 µM $ZnCl_2$ and 40 µg/ml kanamycin for overnight protein expression at room temperature. After expression, membranes were transferred to empty petri dishes and the cells were lysed by freeze-thaw cycles (three times, −20 and 37°C, 10 min). To assay activity, 25 ml of 0.5% Agarose in 50 mM Tris-HCl buffer pH 7.5 containing 200 µM $ZnCl_2$ and 250 µM *p*-nitrophenyl phenylphosphonate (Sigma) was poured onto the membrane. Colonies with active enzymes developed a yellow colour due to the hydrolyzed substrate. The most active colonies (~200 variants) were directly picked from plates for rescreening in 96-well plates.

## Cell lysate activity screen in 96-well plates

To test the PMH fitness and solubility of the variants, individual wells of a 96-well plate containing 400 µl of LB media supplemented with 40 µg/ml kanamycin were inoculated with 20 µl of overnight culture and incubated at 30°C for 3 hr. Protein expression was induced by adding IPTG to a final concentration of 1 mM and further incubation at 30°C (20°C and 37°C for testing temperature effect on expression) for 3 hr. Cells were harvested by centrifugation at 4000 × g for 10 min and pellets were frozen at −80°C for at least 30 min. For lysis, cell pellets were resuspended in 200 µl of lysis buffer (50 mM Tris-HCl pH 7.5, 100 mM NaCl, 200 µM $ZnCl_2$, containing 0.1% Triton X-100, 100 µg/ml lysozyme and 1 U/ml of benzonase) and incubated at 25°C with shaking at 1200 rpm for 1 hr. The cell lysates were clarified by centrifugation at 4000 × g for 20 min at 4°C. Clarified lysates were diluted twofold for phosphonate hydrolase activity prior activity assays in order to obtain linear initial rates, which were measured at a single substrate concentration of 500 µM. 'PMH fitness' is defined

as the rate of product formation (nM/s) measured in cell lysate using an extinction coefficient of 18,300 $M^{-1}$ $cm^{-1}$ (for the p-nitrophenol product) and correcting for lysate dilution (i.e. $\times$ 2 fold in this case).

## Purification of Strep-tagged proteins

All variants were cloned as described above, transformed, overexpressed in *E. coli* BL21 (DE3) cells and purified as described previously (*Baier and Tokuriki, 2014*).

## Enzyme kinetics

The kinetic parameters and activity levels of purified of enzyme variants were obtained as described previously (*Baier and Tokuriki, 2014*). Briefly, phosphonate monoester hydrolase (PMH) hydrolysis was monitored following the release of *p*-nitrophenol at 405 nm with an extinction coefficient of 18,300 $M^{-1}$ $cm^{-1}$. The β-lactamase activity was monitored at 405 nm using Centa as substrate, and molar product formation was calculated with the extinction coefficient of 6300 $M^{-1}$ $cm^{-1}$.

## Thermostability assay

The thermal stability of variants was measured with a thermal shift assay as described previously (*Wyganowski et al., 2013*). Briefly, enzyme variants (2 µM) were mixed with 5 × SYPRO Orange dye (Invitrogen) in a 20 µl reaction and heated from 25°C to 95°C in a 7500 Fast Real-Time PCR system (Applied Biosystems). Measurements were conducted in triplicate and unfolding was followed by measuring the change in fluorescence caused by binding of the dye (excitation, 488 nm; emission, 500–750 nm). The melting temperature ($T_m$) is calculated from midpoint of the denaturation curve and values were averaged.

## Protein purification for crystallization

The NDM1 and VIM2 protein variants were expressed in *E. coli* BL21 (DE3) cells in TB medium (400 ml) supplemented with 1% glycerol, 50 µg/ml kanamycin and 200 µM $ZnCl_2$. Cells were grown at 30°C for 6 hr. The temperature was lowered to 22°C and the cells were incubated for a further 16 hr and harvested by centrifugation for 15 min at 8500 × g (R9A rotor, Hitachi), then resuspended in buffer A (50 mM HEPES pH 7.5, 500 mM NaCl, 20 mM imidazole, and 200 µM $ZnCl_2$) and lysed by sonication (OMNI sonic ruptor 400). Cellular debris was removed by centrifugation at 29,070 × g for 60 min (R15A rotor, Hitachi). The supernatant was loaded onto a 5 ml Ni-NTA superflow cartridge (Qiagen) followed by extensive washing with buffer A prior to elution of proteins in buffer B (50 mM HEPES pH 7.5, 500 mM NaCl, 500 mM imidazole, and 200 µM $ZnCl_2$). Protein containing fractions were analyzed by SDS-PAGE (Bolt Mini Gels, Novex). Buffer B containing the proteins was exchanged to TEV reaction buffer (50 mM Tris-HCl pH 8.0, 0.5 mM EDTA, 1 mM DTT, and 150 mM NaCl) using HiPrep 26/10 desalting column (GE healthcare). 20% TEV (w/w) was added and incubated at 4°C for 4 days. The TEV reaction buffer was exchanged to buffer A before TEV protease and His-tag containing debris were removed by Ni-NTA superflow column (5 ml, Qiagen). His-tag cleaved protein was then concentrated using a10 kDa molecular weight cut-off MWCO ultrafiltration membrane (Amicon, Millipore) and loaded on HiLoad 16/600 Superdex 75 pg column (GE Healthcare). Proteins were eluted into crystallization buffers.

## Crystallization of NDM1-R10

NDM1-R10 protein in crystallization buffer (20 mM HEPES pH 7.5, 2 mM β-mercaptoethanol, 150 mM NaCl, and 100 µM $ZnCl_2$) was concentrated to 15 mg/ml using a 10 kDa molecular weight cut-off ultrafiltration membrane (Amicon, Millipore) and crystallized by the hanging drop method. The hanging drops were prepared by mixing protein solution (1 µl) and well solution (2 µl). Crystals appeared within two weeks in 0.1 M MES (pH 6.75) and 1.3 M $MgSO_4$ at 18°C and continued to grow. Crystals were soaked in a cryoprotectant solution for 30 s (precipitant, and 25% glycerol), and flash cooled in liquid nitrogen. The *apo* (without substrate) enzyme crystal diffracted to 1.67 Å at beam line MX1 at the Australian Synchrotron. The product-bound structure was obtained by soaking the crystal in precipitant solution, containing 15 mM substrate for 3-to-30 minutes before soaking in cryoprotectant solution and flash cooling in liquid nitrogen. The crystals diffracted to 1.68–2 Å at beam line MX2 (0.9537 Å) at the Australian Synchrotron.

## Crystallization of VIM2-R10

The first size exclusion peak of VIM2-R10 protein (dimeric fractions), in buffer containing 50 mM HEPES pH 7.5, 150 mM NaCl, and 200 µM ZnCl$_2$, was concentrated to 2.6 mg/ml using a 10 kDa molecular weight cut-off ultrafiltration membrane (Amicon, Millipore) and crystallized by the hanging drop method. The drops were prepared by mixing a protein solution (2 µl), well solution (4 µl), 1 mM TCEP, and 2.5 mM p-nitrophenyl phenylphosphonate. Crystals appeared after two weeks in 0.1 M HEPES (pH 7.5) and 1.2 M sodium citrate at 18°C. Crystals were soaked in cryo-protectant solution for several minutes (precipitant, and 10% glycerol), and flash cooled in liquid nitrogen. The crystal diffracted to 2 Å at beam line MX1 (0.9537 Å) at the Australian Synchrotron.

## Data collection and structure determination

The crystallographic data were collected at 100 K at the Australian Synchrotron. Data were processed using XDS (*Kabsch, 2010*). Scaling was performed using Aimless in the CCP4 program suite. Resolution estimation and data truncation were performed by using overall half-dataset correlation CC(1/2)>0.5 (*Karplus and Diederichs, 2012*). Molecular replacement was used to solve all structures with MOLREP (*Vagin and Teplyakov, 2010*) using the structures deposited under PDB accession codes 3SPU and 1KO3 as starting models for NDM1 and VIM2, respectively. The model was refined using phenix.refine (*Afonine et al., 2012*) and Refmac v5.7 (*Murshudov et al., 2011*) in CCP4 v6.3 program (*Winn et al., 2011*), and the model was subsequently optimized by iterative model building with the program COOT v0.7 (*Emsley and Cowtan, 2004*). The structure of VIM2-R10 in its *apo* form (without substrate) (PDB-ID: 6BM9), and of NDM1-R10 in its *apo* form (PDB ID: 5JQJ) or in complex with the phenylphosphonic acid product (pdb ID: 5K4M) have been deposited in the RCSB Protein Data Bank, www.rcsb.org. For a complete description of the crystallographic data collection and refinement statistics, see *Supplementary file 3*.

## Molecular dynamics simulation

Molecular dynamics simulations VIM2 and NDM1 variants were performed using the Q simulation package (*Marelius et al., 1998*) and the OPLS-AA force field (*Jorgensen et al., 1996*). OPLS-AA compatible parameters for *p*-nitrophenyl phenylphosphonate (PMH substrate, PPP) were generated using MacroModel version 10.3 (Schrödinger LLC, v. 2014–1). Partial charges for PPP were calculated using the standard RESP procedure (*Cieplak et al., 1995*), with the use of Antechamber (AmberTools 12) (*Wang et al., 2006*) and Gaussian09 (Revision C.01) (*Frisch et al., 2016*). All other PPP parameters were as presented in the supporting information of Ref. (*Barrozo et al., 2015*). The structures of VIM2-WT (Chain B, PDB ID 4PVO) and NDM1-WT (PDB ID 4HL2) were obtained from the Protein Data Bank. These structures were used for MD calculations because of their high-resolution, whereas 3SPU and 1KO3 were used for structural analysis due to the absence of ligand in the active site. The structure of NDM1-R10 with the PMH product, phenylphosphonic acid bound (PDB ID 5K4M) was obtained as described in the previous section. The structures of single W93G mutants for both VIM2 and NDM1 were manually generated by mutating the respective tryptophan residues to glycine in the wild-type structures prior MD simulation. The PMH substrate was manually placed in the active site of the respective enzymes, based on the position of the phenylphosphonic acid product in the NDM1-R10 crystal structure. The Zn$^{2+}$ ions were described using a tetrahedral dummy model based on the dummy model originally described by *Aqvist and Warshel (1989)*. The model was built by placing four dummy atoms in a tetrahedral geometry around a central metal particle, and parametrised to reproduce the experimental solvation free energy and solvation geometry of the zinc ion (for the description of analogous octahedral dummy model and parameterization procedure see Ref. (*Duarte et al., 2014*)). The resulting Zn$^{2+}$ parameters used in this work are presented in *Supplementary file 4A*. All simulations were performed using surface-constrained all-atom solvent (SCAAS) boundary conditions (*King and Warshel, 1989*) applied to a spherical droplet of TIP3P water molecules (*Jorgensen et al., 1983*) with a radius of 24 Å centered on the bridging hydroxide ion. All protein atoms and water molecules within 85% of the solvent sphere were allowed to move freely with no restraints, atoms in the last 15% of the sphere were subject to 10 kcal mol$^{-1}$ Å$^{-2}$ positional restraints, and all atoms outside this sphere were subjected to 200 kcal mol$^{-1}$ Å$^{-2}$ positional restraints to maintain them at their crystallographic positions. Protonation states of all ionisable residues within the inner 85% of the simulation sphere were assigned using PROPKA 3.1

(*Søndergaard et al., 2011*; *Olsson et al., 2011*) and the protonation states of histidine side chains were determined by visual inspection of the surrounding hydrogen bonding pattern of each residue. The relevant protonation states and histidine protonation parameters are presented in *Supplementary file 4B*. All ionizable residues outside of the 85% of the sphere were kept in their uncharged forms to avoid simulation artefacts caused by having charged residues in the excluded region of the simulations. All systems were initially equilibrated with 200 kcal mol$^{-1}$ Å$^{-2}$ positional restraints over the total timescale of 95 ps, during which the alternating heating, cooling and reheating was performed to release steric clashes and equilibrate the positions of the solvent molecules and hydrogen atoms, and to reach the target simulation temperature of 300K. The initial equilibration was completed by performing 10 ns of simulation at 300K, which was followed by 100 ns production simulation, the last 50 ns of which was subject to further analysis. This protocol was repeated five times generating five independent 100ns trajectories for each system. The root mean square deviations of all C-$\alpha$ atoms in our simulations is shown in *Figure 5—figure supplement 2*, which demonstrates that all simulations have equilibrated during the 100 ns simulation time. During the final stage of the equilibration and production simulations weak 0.5 kcal mol$^{-1}$ Å$^{-2}$ restraints were applied on the PMH substrate in order to keep it within the simulation sphere, and 1.0 kcal mol$^{-1}$ Å$^{-2}$ restraints were applied on the metal ions, side chains of the metal ligands and the bridging hydroxide ion to assure proper coordination geometry of the metal centers. Apart from the very initial stages of the equilibration where a 0.01 and 0.1 fs time step was used, a 1 fs time step was used throughout the simulations. The results of the simulations were analyzed using: VMD (*Humphrey et al., 1996*), and GROMACS (*Abraham et al., 2015*). The raw data of the MD simulations are available from the Dryad Digital Repository at doi:10.5061/dryad.qk653b3.

## Acknowledgements

We thank Dan S Tawfik, Amir Aharoni, Joelle Pelletier and the members of the Tokuriki lab for comments on the manuscript. Natural Sciences and Engineering Research Council of Canada (NSERC) Discovery Grant (RGPIN 418262–12 and RGPIN 2017–04909), and Canadian Institute of Health Research (CIHR) Foundation Grant to NT. NT is a CIHR new investigator and a Michael Smith Foundation of Health Research (MSFHR) career investigator. We also thank the Knut and Alice Wallenberg and Wenner-Gren Foundations for fellowships to SCLK and AP respectively, as well as the Swedish National Infrastructure for Computing (SNAC) for supercomputing resources.

## Additional information

### Funding

| Funder | Grant reference number | Author |
|---|---|---|
| Natural Sciences and Engineering Research Council of Canada | RGPIN 418262-12 | Nobuhiko Tokuriki |
| Canadian Institutes of Health Research | 353714 | Nobuhiko Tokuriki |
| Natural Sciences and Engineering Research Council of Canada | RGPIN 2017-04909 | Nobuhiko Tokuriki |

The funders had no role in study design, data collection and interpretation, or the decision to submit the work for publication.

### Author contributions

Florian Baier, Conceptualization, Data curation, Formal analysis, Methodology, Writing—original draft, Conceived and designed this study, Performed experimental evolution, enzymatic assay and mutational analysis; Nansook Hong, Paul D Carr, Data curation, Software, Formal analysis, Collected structural data; Gloria Yang, Charlotte M Miton, Data curation, Formal analysis, Writing—review and editing, Performed experimental evolution, enzymatic assay and mutational analysis; Anna Pabis,

Alexandre Barrozo, Data curation, Software, Formal analysis, Designed and conducted MD simulations; Shina CL Kamerlin, Resources, Software, Supervision, Validation, Writing—review and editing, Designed and conducted MD simulations; Colin J Jackson, Resources, Supervision, Validation, Writing—review and editing, Collected structural data; Nobuhiko Tokuriki, Conceptualization, Resources, Formal analysis, Supervision, Funding acquisition, Writing—original draft, Writing—review and editing, Conceived and designed this study

### Author ORCIDs
Gloria Yang (iD) https://orcid.org/0000-0002-9139-6148
Charlotte M Miton (iD) https://orcid.org/0000-0002-2374-303X
Shina CL Kamerlin (iD) https://orcid.org/0000-0002-3190-1173
Colin J Jackson (iD) https://orcid.org/0000-0001-6150-3822
Nobuhiko Tokuriki (iD) https://orcid.org/0000-0002-8235-1829

### Decision letter and Author response
Decision letter https://doi.org/10.7554/eLife.40789.030
Author response https://doi.org/10.7554/eLife.40789.031

## Additional files

### Supplementary files
• Supplementary file 1. Description of enzymes and directed evolution rounds. (**A**) Information about the enzymes characterized in this study. (**B**). Catalytic parameters of enzymes characterized in this study for PMH and β-lactamase activities. (**C**) Mutation rate in the NDM1-wt and VIM2-wt naive libraries. (**D**) Mutations identified in the naive library of VIM2-wt. (**E**) Mutations identified in the naive library of NDM1-wt. (**F**) Information on the directed evolution procedure and mutations that were accumulated during directed evolution. (**G**) PMH fitness values (cell lysate activity) of evolved and designed mutants. (**H**) Variants identified in the screening of additional round 1 libraries 1–3. (**I**) Changes in catalytic activity of purified MBL mutants compared to their respective wild-type enzymes, and melting temperature.
DOI: https://doi.org/10.7554/eLife.40789.016

• Supplementary file 2. Kinetic parameters of all variants. (**A**) Catalytic parameters of VIM2 variants. (**B**) Individual catalytic parameters of NDM1 variants. (**C**) Solubility and melting temperature of NDM1 and Vim2 variants.
DOI: https://doi.org/10.7554/eLife.40789.017

• Supplementary file 3. Crystallographic information. Crystallographic data collection and refinement statistics.
DOI: https://doi.org/10.7554/eLife.40789.018

• Supplementary file 4. Molecular dynamics information. (**A**) Parameters used to describe the $Zn^{2+}$ ions in our MD simulations. (**B**) List of relevant ionized states as well as the protonation patterns of histidine residues in our molecular dynamics simulations. All other residues were kept in their unionized forms as they were outside the simulation sphere (see main text).
DOI: https://doi.org/10.7554/eLife.40789.019

• Transparent reporting form
DOI: https://doi.org/10.7554/eLife.40789.020

### Data availability
Diffraction data have been deposited in PDB under the accession code 5JQJ, 5K4M and 6BM9.

The following datasets were generated:

| Author(s) | Year | Dataset title | Dataset URL | Database and Identifier |
|---|---|---|---|---|
| Baier F, Hong N, Yang G, Pabis A, Miton CM, Barrozo A, Carr PD, Ka- | 2018 | Data from: Cryptic genetic variation defines the adaptive evolutionary potential of enzymes | http://dx.doi.org/10.5061/dryad.qk653b3 | Dryad Digital Repository, 10.5061/dryad.qk653b3 |

| | | | | | |
|---|---|---|---|---|---|
| merlin SCL, Jackson CJ, Tokuriki N | | | | | |
| Hong N-S, Jackson CJ, Carr PD | 2018 | Directed evolutionary changes in MBL super family - NDM-1 Round 10 crystal-1 | https://www.rcsb.org/structure/5JQJ | | Protein Data Bank, 5JQJ |
| Hong N-S, Jackson CJ, Carr PD | 2018 | Directed evolutionary changes in MBL super family - NDM-1 Round 10 crystal-3 | https://www.rcsb.org/structure/5K4M | | Protein Data Bank, 5K4M |
| Hong N-S, Jackson CJ, Carr PD | 2018 | Directed evolutionary changes in MBL super family - VIM-2 Round 10 | https://www.rcsb.org/structure/6BM9 | | Protein Data Bank, 6BM9 |

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
