## [Decision Letter]

Thank you for submitting your article "Cryptic genetic variation defines the adaptive evolutionary potential of enzymes" for consideration by *eLife*. Your article has been reviewed by three peer reviewers, including Andrei N Lupas as the Reviewing Editor and Reviewer #1, and the evaluation has been overseen by Patricia Wittkopp as the Senior Editor.

The reviewers have discussed the reviews with one another and the Reviewing Editor has drafted this decision to help you prepare a revised submission.

Summary:

This is a carefully executed study of the evolutionary trajectories taken by four metallo-β-lactamases (MBLs) in response to selection for phosphonate monoester hydrolase activity. Supported by an impressive array of data, the authors argue that cryptic properties of the evolutionary starting points of the different MBLs determine their evolvability. The main finding is that the initial promiscuous activities of the WT enzymes are not indicative of the final fitness peaks and that an orthologue with lower initial activity reached the highest fitness plateau in this experiment. The trajectories of two of the four orthologues are tracked in considerable detail and represent a comprehensive exploration of the molecular basis for the emergence of a new enzyme function from a low-level side activity.

Essential revisions:

A major question on which the approach hinges is the reproducibility of the trajectories, an issue addressed in the subsection “A limited accessibility to adaptive mutations from each starting point”. Given the stochastic nature of the starting library, would the evolutionary trajectory for a given protein be the same every time? Surprisingly, the article does not try to replicate the trajectories shown in Figure 1D, in order to show that these remain substantially the same, irrespective of the starting conditions. Instead, it presents additional samplings of starting libraries for two of the proteins (Figure 2A), data which appear to correspond to the R1 generation in Figure 1D. The extent to which these samplings substantiate the deterministic nature of the trajectories is not clear. The other arguments presented to this point seem equally indirect. If this issue cannot be resolved, the claims of determinism throughout the paper will have to be reduced substantially.

The article makes a number of sweeping claims, including that the results reported are essential to predicting the evolution of proteins and that they have profound implications for protein design. However, there is nothing in the paper to indicate that, on the basis of this study, the authors could predict the trajectory of a fifth MBL from its sequence. There are also no clear implications for protein design. The conclusion that multiple starting points are better than a single one refers only to directed evolution studies. In light of this, it is necessary to carefully edit the article in order to remove all claims not borne out by the data.

---

## [Author Response]

Essential revisions:A major question on which the approach hinges is the reproducibility of the trajectories, an issue addressed in the subsection “A limited accessibility to adaptive mutations from each starting point”. Given the stochastic nature of the starting library, would the evolutionary trajectory for a given protein be the same every time? Surprisingly, the article does not try to replicate the trajectories shown in Figure 1D, in order to show that these remain substantially the same, irrespective of the starting conditions. Instead, it presents additional samplings of starting libraries for two of the proteins (Figure 2A), data which appear to correspond to the R1 generation in Figure 1D. The extent to which these samplings substantiate the deterministic nature of the trajectories is not clear. The other arguments presented to this point seem equally indirect. If this issue cannot be resolved, the claims of determinism throughout the paper will have to be reduced substantially.

We thank the reviewers for their comments. We agree that the repeatability and deterministic nature of these four evolutionary trajectories are two aspects that are strongly associated with our overall claim. Studying these questions at the genetic level using our system, however, is beyond our experimental capacities. The main reason for not replicating the long-term trajectories of VIM2 and NDM1, with multiple parallel lineages, is the intensity of the experimental scheme required to perform directed evolution. Unlike other studies, which employ in vivo enzyme selection and organismal evolution, our directed evolution scheme (library creation, and cell lysate screenings) requires several weeks per round and per enzyme. Obtaining further mutational analyses, kinetics and structures can take many additional months. Thus, performing such analysis in multiple repeated lineages for multiple starting points is virtually unfeasible. Consequently, we conducted the most informative approach, within our reach, to perform a re-screening of three additional round 1 libraries for both NDM1 and VIM2 to probe the extent of determinism, at least at the initial round. The experiment showed striking repeatability and occurrence of key mutations that confer the largest increase of the evolutionary trajectories in the additional screenings (Figure 2). These results suggest that only a handful of adaptive mutations that are unique to each genotype, are accessible, at least from the wild-type sequence. However, we agree that this experiment does not prove the repeatability of the evolutionary outcomes, and other evidence and arguments that we presented in the manuscript are still indirect and circumstantial.

Therefore, in agreement with the reviewers, we have substantially reduced our claim of determinism throughout the manuscript; in particular, in the section “A limited accessibility of adaptive mutations from each starting point”, which has been extensively revised to comply with the reviewers’ comments.

The article makes a number of sweeping claims, including that the results reported are essential to predicting the evolution of proteins and that they have profound implications for protein design. However, there is nothing in the paper to indicate that, on the basis of this study, the authors could predict the trajectory of a fifth MBL from its sequence. There are also no clear implications for protein design. The conclusion that multiple starting points are better than a single one refers only to directed evolution studies. In light of this, it is necessary to carefully edit the article in order to remove all claims not borne out by the data.

We thank the reviewers for this comment and have edited the Discussion accordingly. However, we would like to clarify that our intention was not to claim that this work will allow us to predict evolutionary outcomes starting from other MBL orthologues. On the contrary, we aim at emphasizing how difficult and complicated it is to predict evolutionary outcomes from a given starting point, partly due to the existence of cryptic genetic variation. We would like to discuss that only a very deep understanding of the molecular mechanisms at play during functional transitions will empower our ability to predict evolution. We also believe that these results are relevant, not only to the field of directed evolution and engineering, but also to protein design because most protein/enzyme designs still rely on a unique starting point, rather than exploring multiple ones. However, we acknowledge that there are researchers, in particular groups belonging to the Rosetta community, who capitalize on multiple starting points, which is why we now cite their work in the Discussion.